# Chemical Characterization of Bioactive Compounds in Extracts and Fractions from *Litopenaeus vannamei* Muscle

**DOI:** 10.3390/md23020059

**Published:** 2025-01-27

**Authors:** Sandra Carolina De La Reé-Rodríguez, María Jesús González, Ingrid Fernández, José Luis Garrido, Erika Silva-Campa, Norma Violeta Parra-Vergara, Carmen María López-Saiz, Isabel Medina

**Affiliations:** 1Departamento de Investigación y Posgrado en Alimentos, Universidad de Sonora, Hermosillo 83000, Sonora, Mexico; a211204990@unison.mx (S.C.D.L.R.-R.); violeta.parra@unison.mx (N.V.P.-V.); 2Química de Productos Marinos, Instituto de Investigaciones Marinas, 36208 Vigo, Pontevedra, Spain; mjgp@iim.csic.es (M.J.G.); ifernandez@iim.csic.es (I.F.); garrido@iim.csic.es (J.L.G.); 3Departamento de Investigación en Física, Universidad de Sonora, Hermosillo 83000, Sonora, Mexico; erika.silva@unison.mx

**Keywords:** *L. vannamei* muscle, antiproliferative, breast cancer, lipidic compounds, EPA and DHA, astaxanthin

## Abstract

Marine organisms are a vital source of biologically active compounds. Organic extracts from the muscle of the Pacific white shrimp (*L. vannamei*) have shown antiproliferative effects on tumor cells, including breast adenocarcinoma. This study aimed to analyze these extracts’ composition and confirm their specificity for breast adenocarcinoma cells without harming normal cells. An organic chloroform extract from *L. vannamei* muscle was divided using a solvent partition procedure with methanol and hexane. The methanolic partition was fractionated through an open preparative liquid chromatography column to isolate compounds with biological activity, that were later tested on MDA-MB-231 (breast adenocarcinoma), and recently tested on MCF10-A (non-cancerous breast epithelial cells). Cells incubated with these fractions were assessed for viability and morphological changes using fluorescence confocal microscopy. Fractions F#13 and F#14 reduced MDA-MB-231 cancer cell viability at 100 µg/mL without affecting non-cancerous MCF-10A cells, inducing apoptosis-related changes in cancer cells. These fractions contained EPA and DHA free fatty acids, specifically F#13 contained free and esterified astaxanthin as well. The high levels of free linoleic acid 18:2 ω-6, EPA, and DHA (in a 2:1 ratio, EPA:DHA), along with free and esterified astaxanthin in F#13, significantly reduced breast adenocarcinoma cell viability, nearly to that achieved by cisplatin, a chemotherapy drug.

## 1. Introduction

White shrimp (*Litopenaeus vannamei*, Boone, 1931), which is native to the eastern Pacific starting from the Gulf of California, México to Tumbes, Northern Perú [1], constitutes an important species for the aquaculture industry, being the most produced crustacean worldwide with 5812.2 million tons [2]. There is evidence that its muscle is a source of compounds with some biological activities such as antioxidant [3], antimutagenic [4], and antiproliferative activity [5]. Organic extracts resulting from the white muscle of *L. vannamei* have recently shown biological activity for inhibiting the cell viability of a number of cancerous cells [4,5,6]. These studies have suggested a potential role of lipids and carotenoid pigments, which could be contained in these biologically active extracts [6]. 

In this field, marine lipid and lipid classes with anti-tumor properties have been identified in recent years [7]. The use of omega-3 polyunsaturated fatty acids (n-3 PUFAs) from marine sources, including eicosapentaenoic acid (EPA) and docosahexaenoic acid (DHA), is suggested in a therapeutic context in patients receiving treatment for a variety of cancer types [8]. Research points out that marine n-3 PUFAs have potential as an effective adjuvant to chemotherapy treatment. Animal and cell culture research suggests that ω-3 PUFAs exert an inhibitory role on cell proliferation and induce programmed cell death (apoptosis) [9]. They can regulate inflammatory processes, cell signaling, gene expression, and lipid metabolism. These roles, together with their anti-inflammatory and antioxidant properties, may address preventive effects, contributing to reducing some of the secondary complications associated with cancer [10]. Intervention studies have also demonstrated the potential link between a higher consumption of dietary marine n-3 PUFAs and a lower risk of breast cancer [11]. Therefore, marine n-3 PUFA studies have been relevant in the pharmacological field, and they are being considered for their therapeutic application.

Additionally, marine lipids and pigments as carotenoids have also been suggested for therapy strategies. Marine carotenoids are generally biosynthesized by autotrophic organisms, such as bacteria and archaea, algae, and fungi. Some heterotrophic organisms (fish, crustaceans) also contain carotenoids, probably accumulated from food or partially modified through metabolic reactions [12]. Moreover, carotenoids are lipid-soluble pigments that have been suggested to be part of biologically active *L. vannamei* extracts [6]. Research has reported that marine carotenoids such as fucoxanthin, astaxanthin, and zeaxanthin can exert strong antioxidant, antiproliferative, and anti-inflammatory effects, being able to inhibit tumor cell proliferation [13]. They are suggested to contribute to inducing apoptosis and cell cycle arrest, helping to prevent cell migration and invasion in different cancer cell lines [13]. Carotenoids can act as chemosensitizers; when combined with conventional anticancer drugs, they can prevent mechanisms of multidrug resistance. They can help to restore the sensitivity of tumors to chemotherapy [14].

This work aims to test the biological activity of marine organic extracts coming from *L. vannamei* muscle, which has previously shown a certain antiproliferative activity over breast carcinoma cells and their innocuity over normal cells as breast epithelial cells. The study intends to explore deeply the composition of these biologically active extracts, confirming their specificity for breast adenocarcinoma cells and identifying the compounds responsible for the biological protective effect.

## 2. Results

### 2.1. Characterization of Lipid Compounds from Chloroform, Hexane, and Methanol Extracts

The lipid classes contained in the initial organic chloroform extract and the subsequent hexane and methanol extracts were quantified. According to the procedure described in the methodology section, 10 mL of chloroform extract provided a yield of 9.7 ± 0.05 mg of dry residue. This initial chloroform extract was then separated into two phases: one soluble in hexane, which provided 2.29 ± 0.12 mg, and one soluble in methanol, which provided 7.38 ± 0.14 mg. These results indicated that approximately 75% of the compounds were concentrated in the methanolic phase. Table 1 shows the lipid composition of these extracts determined by TLC (Figure 1). The initial chloroform extract contained around 43% of neutral lipids and 57% of polar lipids, mainly phosphatidylcholine (PC) and phosphatidylethanolamine PE. Non-polar lipids, as free fatty acids, and cholesterol were then concentrated in the hexane phase, and polar lipids were concentrated in the methanolic phase. It can be observed that, within neutral lipids, the concentration of cholesterol is higher, followed by free fatty acids. As for polar lipids, the majority is phosphatidylethanolamine.

Hexane extract turned out to be rich in cholesterol and had a low concentration of polar lipids (Figure 1), and methanolic extract concentrated the polar lipids, specifically phosphatidylethanolamine (Table 1 and Figure 2), having low amounts of mono- and triacyl-glycerides with concentrations below the quantification limits. This means that the methanol phase, which was fractionated, contained polar lipids and no apolar lipids.

### 2.2. Characterization of Lipid Compounds from Isolated Column Fractions Coming from the Methanol Extract

Figure 3 shows the lipid classes TLC analysis of the isolates coming from the methanolic extract fractionation through the column. The elution protocol was followed according to previous works [5,6] and provided 22 isolated fractions. TLC results demonstrated that the lipid compounds were contained in fractions numbered 3–4 (F#3 and F#4), fractions numbered 10–16 (F#10–F#16), and in fraction 21 (F#21). Cholesterol was found in fractions 3–4. Eluates from F#10 to F#16 contained a variety of free fatty acids. Fraction 21, recovered with 100% methanol, contained the polar lipids. According to previous results executed in human cancer cell lines [6], fractions F#13 and F#14 were those that previously caused a significant decrease in the viability of a breast adenocarcinoma cell line (MDA-MB-231), meaning they have antiproliferative potential [6]. They were formed by free fatty acids. 

### 2.3. Fatty Acid Composition of Chloroform, Hexane, and Methanol Extracts, and Isolated Fractions F#13 and F#14

Table 2 shows the fatty acid composition of lipids present in the chloroform extract, hexane, and methanol phases. The data of chloroform extract show a higher content of polyunsaturated fatty acids (PUFA), mainly 18:2 n-6 and 20:5n-3 and 22:6n-3 as n-3 PUFA, than saturated (SAT) and monounsaturated fatty acids (MUFA). Fractionation into hexane and methanol provided a higher concentration of PUFA ω3 in the methanolic extract in agreement with its high proportion of polar lipids. 

A high percentage of EPA and DHA was detected in fractions F#13 and F#14 (Table 3). According to the data, F#13 leads to higher amounts of 18:2n-6 and EPA than F#14. On the other hand, the concentration of n-3 polyunsaturated fatty acids was higher in F#14, specifically docosahexaenoic acid (DHA) followed by EPA.

### 2.4. Identification and Quantification of Pigments

The presence of astaxanthin (AX) and beta carotene was confirmed in chloroform, hexane, and methanol extracts (Table 4). Chloroform extract contained a higher amount of AX compounds than beta carotene. The subsequent fractionation provided a methanolic extract, in which the AX content was concentrated, and a hexane extract, which concentrated the high amount of beta carotene according to the pigment solubility in both solvents. 

In F#13, it was possible to elucidate the presence of astaxanthin, mostly in its free form (Table 5), and mono- and di-esterified polyunsaturated, monounsaturated, and saturated fatty acids were also found. However, the pigments in F#14 were below the detection limit.

### 2.5. Identification and Quantification of Tocopherol

Tocopherol isomers were presented in low concentrations in the chloroform and in the methanol extracts (Table 6), and they were not detected in the hexane extract. The alfa, beta, and gamma isomers of tocopherol were identified, with the alpha-tocopherol being the major isomer. As regards the fractions eluted from the column, tocopherol traces were detected in the first fractions until fraction number 3. It is important to note that after fraction 3, the tocopherol level decreased to the extent that the presence of tocopherols in the fractions with biological activity was not identified.

### 2.6. MDA-MB-231 and MCF-10A Viability

Previous studies in colon carcinoma (HCT-116), breast adenocarcinoma (MDA-MB-231), and retinal pigment epithelial (ARPE-19) cell lines have pointed to a potential biological effect of specific fractions coming from the extraction protocol from *L. vannamei* muscle here described [6]. Such fractions are those corresponding to eluates fractions numbered 13 (F#13) and 14 (F#14). Viability assays were focused on the activity of those fractions numbered 13 (F#13) and 14 (F#14). 

Figure 4 and Figure 5 show the dose-dependent effect of F#13 and F#14 for MDA-MB-231 and MCF-10A cell viability, respectively. Cisplatin and doxorubicin, as effective chemotherapy compounds in the treatment of several cell carcinomas, were used as references. Also, EPA as n-3 PUFA, present in both isolated fractions, was also added to the cells as a control. 

Data in Figure 4 demonstrated an F13 dose-time dependent effect for the highest concentrations in the MDA-MB-231 cell line, showing an almost 50% reduction of cell metabolism when applied at 100 µg/mL (viability value: 50.98 ± 5.12%). Concentrations ranging between 50 and 12.5 µg/mL did not show any effect. Incubation with F#14 applied at 100 µg/mL was effective in reducing cell viability by around 30% as well (viability value: 68.43 ± 5.96), and there was no effect for concentrations ranging between 50 and 12.5 µg/mL. As regards controls, chemotherapy compounds doxorubicin and cisplatin added at 100 µg/mL inhibited the viability of cancerous MDA-MB-231 cells (around 80% and 65%, respectively). EPA was effective in inhibiting cell viability when employed at 100 µg/mL, rending significantly lower viability percentages (12.65 ± 3.96%) compared to fractions #13 and #14.

On the other hand, Figure 5 shows that the viability of MCF-10A was not significantly affected by the isolated fractions F#13 and F#14. Indeed, these results are supported by Figure 6, where cell morphology can be observed on brightfield after 48 h of treatment. It is observed that the morphology of the cells treated with F#13 and F#14 is not affected; some cells are even in the process of division. On the other hand, it is noted that cells treated with EPA at 100 µg/mL exhibit the characteristic morphology of dead cells, similar to those treated with cisplatin and doxorubicin. Both these drugs and EPA do not show selectivity for breast cancer cells and can cause damage to non-cancerous breast tissues. It is important to reiterate that the combination of n-3 PUFA fatty acids and free and esterified astaxanthin found in F#13 acts selectively against cancer cells without affecting non-cancerous cells.

### 2.7. Morphological Changes

Double staining with DAPI and Phalloidin was used to observe the MDA-MB-231 nucleus and cytoskeleton morphology respectively (Figure 7). The first image (Figure 7.1 control) shows a cell’s natural morphology; however, cells incubated with fractions F#13 and F#14, as well as the other controls, showed morphological changes associated with apoptosis, such as a reduction in cytoplasmic volume and irregular shapes, plasma membrane blebs, and chromatin condensation (piknosis). However, it is necessary to perform quantitative tests to determine the mechanism of cell death caused by fractions 13 and 14, whether by apoptosis or necrosis.

## 3. Discussion

According to the results obtained by thin-layer chromatography, the crude extract and the methanolic phase are rich in polar lipids, which is consistent with a previous study [15] where muscle extracts of the same species contained approximately 77% of polar lipids. On the other hand, fractions 13 and 14 are rich in polyunsaturated fatty acids from the omega-3 family, mainly EPA and DHA, which coincide with the fractions that have antiproliferative potential on MDA-MB-231(breast adenocarcinoma) in a previous study [6]. EPA and DHA are known to present benefits to human health, such as in the reduction of cardiovascular and neurodegenerative diseases [16].

Fraction 13 contains astaxanthin (Table 5), mostly in its free form, but also mono- and di-esterified with polyunsaturated fatty acids (EPA and DHA). In a previous study, purified free astaxanthin was tested and found to have no antiproliferative potential on MDA-MB-231 [6]. However, it seems interesting to conduct studies to identify the role that esterified astaxanthin plays in the processes of breast adenocarcinoma cell death.

Astaxanthin possesses strong activity against free radicals and other prooxidant molecules. This effect is due to its molecular structure (characterized by polar ion rings and non-polar carbon-carbon conjugate bonds), which confers an antioxidant property 10 times greater than other carotenoids, such as lutein, canthaxanthin, and β-carotene [17]. As is well known, lipids are highly vulnerable to reactive oxygen species. Therefore, astaxanthin, being esterified to PUFAs, could be helping to prevent the oxidation of unsaturated fatty acids [18,19] and, therefore, contribute synergistically to the biological potential of F#13, whereas, in F14, the astaxanthin presence could not be identified since its concentration was below the detection limit (Table 5). It is important to mention that the low concentration of pigments may be related to the amount of sample with which extractions were made and that the presence or absence of one or the other carotenoid could depend on the food and culture conditions of shrimp farms.

Tocopherols are antioxidants that react with peroxide radicals and inactivate them; in nature, they can be found in several isomers, including α, β, γ, and δ tocopherol [20]. These three isomers are found in the crude extract, in the methanolic phase, and in the first three fractions. From the fourth fraction onwards, none of the three isomers are detected. Therefore, fractions 13 and 14 do not contain tocopherol, and the antiproliferative potential cannot be attributed to this compound.

F#13 at 100 µg/mL showed significantly lower viability percentages compared to the other treatments, followed by F#14 at the same concentration. This event could be related to this difference in fatty acids and their proportions. EPA and DHA are known for their great benefits to human health. Some studies relate the consumption of EPA and DHA with reduced risk of cardiovascular diseases, diabetes, and cancer, among others [16]. It has also been studied that these fatty acids can regulate inflammatory processes, cell signaling, gene expression, lipid metabolism, and metabolic diseases; therefore, their study has been relevant in the pharmacological field and has been considered for its therapeutic application [12].

Furthermore, it has been reported that n-3 fatty acids, such as those that are EPA and DHA purified, demonstrate antiproliferative potential in MDA-MB-231 by activating signaling pathways involucrate with p38-MAPK, JNK, and ERK proteins, as well as reducing the expression of integrin β3, a protein that promotes metastasis [21]. Therefore, these fatty acids could be considered responsible for the potential antiproliferative; however, it is important to elucidate how these polyunsaturated fatty acids interact with the other compounds found in the fractions, such as astaxanthin, and how they influence the development of MDA-MB-231. On the other hand, non-cancerous breast cells were not affected by fractions 13 and 14. These results may indicate a level of selectivity of the fractions 13 and 14 towards membrane receptors or molecules present in breast cancer cells (MDA-MB-231). 

The images captured using confocal fluorescence microscopy suggest that breast adenocarcinoma cells (MDA-MB-231) treated with fractions 13 and 14 show changes related to cell death by apoptosis. Some of the observed changes are chromatin condensation or pyknosis, an alteration that has been associated with apoptosis [22], and changes in the cytoskeleton were observed, including irregular shapes such as microtubule peaks and rounded cells with membrane blebs, which are also related to apoptosis. The presence of each cytoskeleton shape may be related to the phase of cell death, which can be in either early or late apoptosis [23]. However, quantitative methods, such as flow cytometry, are necessary to understand how fractions 13 and 14 are involved in the cell death mechanism or how they intervene in the cell cycle of MDA-MB-231. As well, it is considered important to conduct more in-depth tests to understand the mechanism of action of the compounds found in these fractions on MDA-MB-231 and MCF10-A, such as Western blot and qPCR, among others, which can provide information about the proteins and genes involved in the process of cell death.

## 4. Materials and Methods

Materials: Frozen shrimp (*Litopenaeus vannamei*) were supplied by a local market (Vigo, Spain). All solvents used were either analytical or HPLC-MS-grade (Merck KGaA, Darmstadt, Germany). Cholesterol, 1,2,3-trihexadecanoylglycerol (TG standard), 1,2-dihexadecanoyl-3-glycerol (DG standard), 1-hexadecanoyl-2,3-glycerol (MG standard), 1,2-dihexadecanoyl-sn-glycero-3-phosphocholine (PC), 1,2-dioctadecanoyl-sn-glycero-3-phosphoethanolamina (PE), and oleic acid were acquired from Larodan (Larodan, Stockholm Sweden). Copper sulfate (II), potassium chloride, nonadecanoic acid, and α, γ, and δ-tocopherol isomers were purchased from Merck (Merck KGaA, Darmstadt, Germany). Alson astaxanthin and β-carotene were purchased from Merck (Merck KGaA, Darmstadt, Germany). 

### 4.1. Extraction of L. vannamei Muscle Compounds

Organic extracts containing compounds having biological activity from *L. vannamei* muscle were isolated according to the methodology described by López-Saiz [5] with some modifications. In a first step, an organic chloroform extract was obtained from the shrimp muscle as follows: 10 g of muscle were homogenized with 50 mL of chloroform (1:5 p:v) and left in constant agitation for 48 h in a cold chamber (4 °C) in darkness. Then, the mixture was centrifuged at 2851× *g* at 4 °C for 10 min, and the organic liquid extract was separated from the rest of the muscle. This crude chloroform extract was dried with nitrogen, and the dry product was recovered with a mixture of equal parts methanol and hexane (2 mL). This mixture was then centrifuged at 2851× *g* at 4 °C, for 10 min and left to separate overnight at −20 °C. The methanol and hexane phases were finally separated and quantified by gravimetry to calculate their concentration. During the handling and processing of *L. vannamei* muscle samples, no melanosis-inhibiting blends such as sulfites or resorcinol were added. 

### 4.2. Fractionation and Isolation of Methanolic Soluble Compounds by Open Column Gravity Chromatography 

Compounds contained into the methanolic phase were separated into a liquid chromatography column of 20 cm length × 1 cm diameter, using silica gel (70–230 Mesh, Sigma-Aldrich, Darmstadt, Germany) as stationary phase 4 mg of the methanol extract were fractionated through an elution sequence with mixtures of hexane and acetone in different proportions, as presented in Table 7.

Once all eluates were recovered, they were dried and recovered with hexane and acetone (1:1) and injected into the TLC eluted with hexane and acetone (80:20) to identify and separate the fractions of interest.

### 4.3. Lipid Composition of the Crude Extracts and Isolated Open Column Fractions

Neutral lipid classes of initial chloroform, methanol, and hexane extracts and isolated fractions coming from the column separation were analyzed by Thin Layer Chromatography (TLC) on Silica Gel 60 25 Glass plates 20 × 20 cm, Merck, according to Christie, 1982 [24]. Prior to use, the silica plates were washed with methanol and activated at 120 °C. The eluent used to carry out the separation of the lipid classes consisted of a mixture of hexane:ether:acetic acid (80:20:2 proportion). Stains were obtained by spraying the plates with a solution of copper sulfate (II) (10%) in phosphoric acid (8%) and heating the plates at 170 °C. Quantification was performed using standard calibration graphs, having concentration values (R1–R4) ranging between: Cholesterol (CHL): 2.69–16.16 µg/µL; Free fatty acids (FFA): 2.69–13.76 µg/µL; Triacyl glycerides (TG): 0.54–5.38 µg/µL; Diacylglycerides (DG): 0.56–5.59 µg/µL; and Monoacylglycerides (MG): 0.44–4.40 µg/µL.

Polar lipids were determined by High-Performance Thin Layer Chromatography (HPTLC) according to Hedegaard and Jensen [25]. Prior to use, the HPTLC plates (Silica gel 60 F254 with concentrating zone 20 × 2.5 cm) were washed with MeOH and activated at 120 °C. The eluent used to carry out the separation of the lipid classes consisted of a mixture of chloroform:methanol:2-isopropanol:ethyl acetate:potassium chloride 0.25% (30:9:25:18:6). A solution of copper sulfate (II) (10%) in phosphoric acid (8%) was also used as spry reagent and stains were revealed by heating the plates at 170 °C. Quantification was performed using standard calibration graphs having concentration values (R1–R4) ranging between: Phosphatidylcholine (PC): 0.73–6.35 µg/µL and Phosphatidylethanolamine (PE): 1.41–8.78 µg/µL. Data were expressed in percentage of dry residue and are the mean standard deviations of two different determinations.

### 4.4. Fatty Acid Analysis

The fatty acid composition of initial chloroform, methanol, and hexane extracts and the interest isolated fractions coming from the column separation were determined by gas chromatography [24]. The lipids were previously derivatized following the methodology of Lepage and Roy, 1986 [26], with a brief modification using a solution of sulfuric acid in methanol.

They were analyzed via Gas chromatography with Flame Ionization Detector equipment (GC/FID, Clarus 500, Perkin–Elmer, Cambridge, MA, USA), using nonadecanoic acid (19:0) as the internal standard. The separation was achieved into a SP^®^-2330 capillary GC column (L × I.D. 30 m × 0.25 mm, d_f_ 0.20 μm, Supelco, Bellefonte, PA, USA) operating at injector and detector temperatures of 275 °C and 260 °C, respectively. The identification of fatty acids was carried out by comparing the retention times with those corresponding to a commercial mixture of fatty acid methyl esters (FAME Mix, Supelco). Data were expressed as percentages of fatty acids related to total fatty acids.

### 4.5. Pigments Analysis

Carotenoid pigments were determined on the initial chloroform, methanol, and hexane extracts and isolated fractions, according to the method by Zapata and Garrido, 2000 [27], with some modifications [28]. Briefly, pigments were analyzed by HPLC (Waters Alliance separation module 2690 coupled to a diode array detector 996, Waters Corporation, Milford, MA, USA) into a Symmetry Waters C8 Column (150 × 4.6 mm, 3.5 µm particle size, 100 Å pore size). Gradient elution was performed with a mixture of solvents in different proportions (A) methanol, acetonitrile, pyridine solution (0.25 M) (50:25:25 *v*:*v*:*v*) and (B) methanol, acetonitrile, acetone (20:60:20 *v*/*v*/*v*) with a flow rate of 1 mL/min for 60 min. Data were expressed as a percentage of pigment related to total pigments of chloroform extract, hexane, and methanol partition and, F#13 and F1#4 fractions.

### 4.6. Tocopherol Analysis

The tocopherol content of the initial chloroform, methanol, and hexane extracts and isolated fractions was determined by HPLC-FL (Alliance Separation Module 2605, coupled with a fluorescence detector, Multi λ Fluorescence Detector 2475, from Waters). Tocopherol was analyzed through an XBridge© Waters C18 column (250 × 4.6 mm, 5 µm particle size, 100 Å pore size). Gradient elution was performed using MeOH (A) and 2-Isopropanol (B)at a flow rate of 1ml/min for 10 min. Fluorescence Detector conditions: λex = 292 nm, λem = 328 nm. The fractions were analyzed directly by taking a volume of them, evaporating the solvent with nitrogen, and resuspending in absolute EtOH. Data were expressed as weight percent of tocopherol of residue in the extracts or isolated fractions.

### 4.7. Cell Viability Analysis

The cell lines MDA-MB-231 (breast adenocarcinoma) and MCF10A (non-cancerous breast epithelial cells) from American Type Tissue Collection (ATTC, Rockville, MD, USA) were grown in Dulbecco’s Modified Eagle Medium (DMEM) in a humidified atmosphere at 37 °C, 5% CO_2_, and 80% humidity.

The standard MTT viability test was performed [29] to determine the effect of isolated fractions on the viability of the MDA-MB-231 and MCF10A cell lines. Briefly, 10,000 cells per well were seeded in 96-well plates, treated with 100, 50, 25, 12.5, and 6.5 μg/mL of each fraction after 24 h, then incubated for 48 h. All test fractions were dissolved in DMSO (Dimethyl Sulfoxide, D2650 Sigma-Aldrich, Darmstadt, Germany) at a maximum concentration of 0.5%, and the controls for cell proliferation (100% growth) were prepared using the same DMSO concentration as in the test fractions. Eicosapentaenoic acid (EPA) (cis-5,8,11,14,17-eicosapentaenoic acid, 20:5 ω-3, E2011 Sigma-Aldrich), cisplatin (15663-27-1 Cis-diaminodicloroplatin II, Sigma-Aldrich), and doxorubicin (D2975000 Doxorubicin hydrochloride, European Pharmacopoeia Reference Standard) were used as controls. 

Cell viability was expressed as a percentage of the control (DMSO-treated cells, set at 100% viability). After 48 h, tetrazolium salt (MTT) was added at a concentration of 5 mg/mL, and optical density readings were taken using a microplate reader (800 TS, BioTek Instruments, Inc., Winooski, VT, USA) at a wavelength of 570 nm (test) and 630 nm (reference). Results were expressed as a viability percentage and mean ± standard deviation (SD) was calculated from three independent experiments, each performed in triplicate.

### 4.8. Cellular Morphology Changes by Fluorescence Confocal Microscopy

The effect of treatment on the morphology and structure of MDA-MB-231 cells was followed according to the method by Van Vuuren et al. (2019) [30]. Cells were seeded in a 6-well plate and incubated for 24 h, then treated with the isolated fractions at the mean inhibitory concentration (IC_50_, 102.04 and 139.14 μg/mL, for fractions numbers 13 and 14, respectively). After 24 h, cells were fixed with 3.7% formaldehyde for 15 min and cell membranes permeated with 0.2% Triton X-100 for 15 min. Cells were stained with 50 μg/mL phalloidin-tetramethylrhodamine B isothiocyanate (phalloidin) (Sigma-Aldrich, MFCD00278840) solution for visualizing F-actin and with 1.5 μg/mL of 4, 6-Diamidine-2-phenyindole, dilactate (DAPI) (Sigma-Aldrich, D9564) solution for staining the genetic (DNA). Images were captured with a Nikon C2+ confocal fluorescence microscope (Nikon Corporation, Tokyo, Japan) and processed with the Nis Elements Nikon 4.30 program to determine fluorescence intensity.

### 4.9. Statistical Analysis

The concentration of lipid classes and fatty acids in the chloroform extract and the hexane and methanol phases was determined by calculating the mean and standard deviation of two different determinations.

Cell viability trials were analyzed implementing one-way ANOVA and a Tukey-HSD test to distinguish means among the different concentrations evaluated for each treatment. These analyses were carried out using Minitab 17 statistical software. IC_50_ values were obtained using probit analysis with the XLSTAT statistical program.

## 5. Conclusions

According to the characterization of extracts and fractions with biological activity obtained from *L. vannamei* muscle, it was observed that chloroform extract is composed of lipids, carotenoids, and tocopherol. The partition of hexane and methanol provided a methanolic extract rich in astaxanthin, polar lipids, and free fatty acids. The fractionation of this methanolic extract resulted in two biologically active fractions labeled as F#13 and F#14, which were able to reduce the viability of MDA-MB-231 cancer cells at 100 µg/mL without affecting the viability of non-cancerous MCF-10A cells. They also induce apoptosis-related morphological changes in cancer cells. Both fractions were composed by EPA and DHA in the form of free fatty acids, and F#13 contained free and esterified astaxanthin as well. In F#13, the presence of 18:2 n-6 (around 25% of total fatty acids) and EPA and DHA in 2:1 proportion present 30% of total fatty acids, together to free and esterified astaxanthin, provided a reduction of breast adenocarcinoma cell viability nearly to that achieved by cisplatin, a chemotherapy compound in the treatment of several cell carcinomas. The results of this work demonstrate that fractions 13 and 14 possess selective antiproliferative potential by decreasing the viability of the breast adenocarcinoma cell line without affecting the viability of breast epithelial cells, which is the desired characteristic when evaluating potential antiproliferative drugs. However, it is necessary to conduct more in-depth tests to understand the mechanism of action of the compounds found in F#13 and F#14 on MDA-MB-231 and MCF10-A, such as flow cytometry, caspase activity, and Western blot, among others, which can provide information about the molecules involved in the process of cell death.

## Figures and Tables

**Figure 1 marinedrugs-23-00059-f001:**
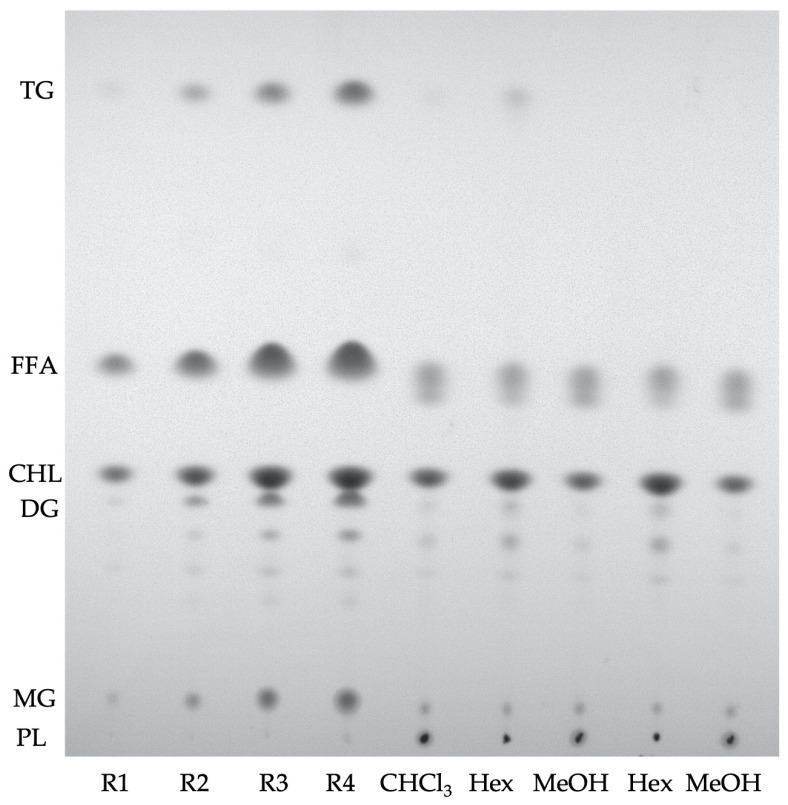
Neutral lipids TLC; R1–R4 standard calibration graph as described in the experimental section; Triacyl glycerides (TG), free fatty acids (FFA), cholesterol (CHL), diacylglycerides (DG), monoacylglycerides (MG), and polar lipids (PL); Chloroform extract (CHCl_3_), hexane partition (Hex), methanol partition (MeOH).

**Figure 2 marinedrugs-23-00059-f002:**
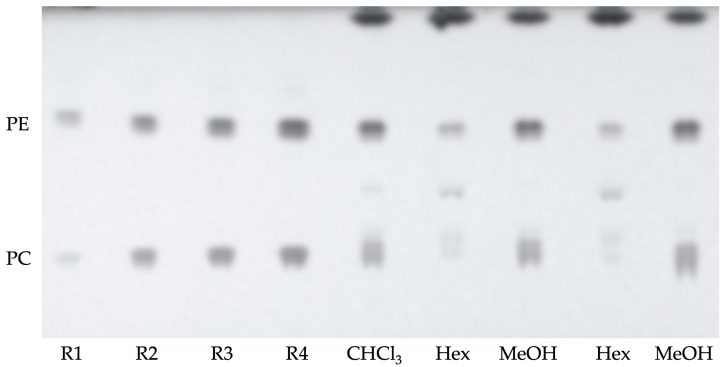
Polar lipids TLC; R1–R4 standard calibration graph as described in the experimental section; Phosphatidylethanolamine (PE) and phosphatidylcholine (PC). Chloroform extract (CHCl_3_), hexane partition (Hex), methanol partition (MeOH).

**Figure 3 marinedrugs-23-00059-f003:**
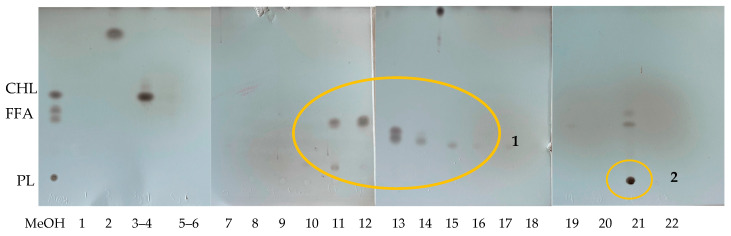
TLC of the 22 eluates obtained from the chromatographic column. 1: The eluates within the circle are the fractions of interest, 2: Polar lipids present in the eluate 21.

**Figure 4 marinedrugs-23-00059-f004:**
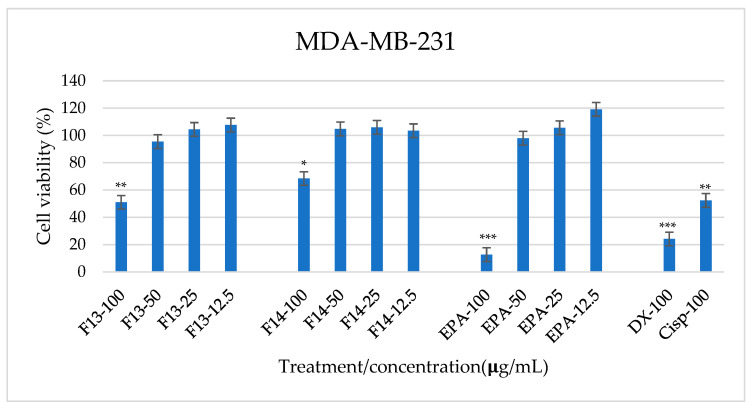
Viability percentage of MDA-MB-231 cells treated with F13 and F14 at different concentrations, as controls eicosapentaenoic acid (EPA), doxorubicin (DX) and, cisplatin (Cisp). Values are mean ± standard error from three independent experiments. Values with different number of asterisks are significantly different (*p* ≤ 0.05). *** significantly lower viability, ** significantly low viability, * significantly different viability.

**Figure 5 marinedrugs-23-00059-f005:**
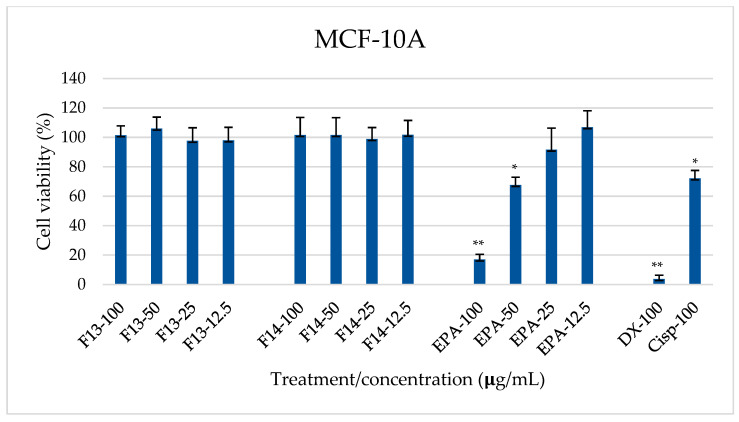
Viability percentage of MCF-10A cells treated with F13 and F14 at different concentrations, as controls eicosapentaenoic acid (EPA), doxorubicin (DX), and cisplatin (Cisp). Values are mean ± standard error from three independent experiments. Values with different number of asterisks are significantly different (*p* ≤ 0.05). ** significantly lower viability and * significantly different viability.

**Figure 6 marinedrugs-23-00059-f006:**
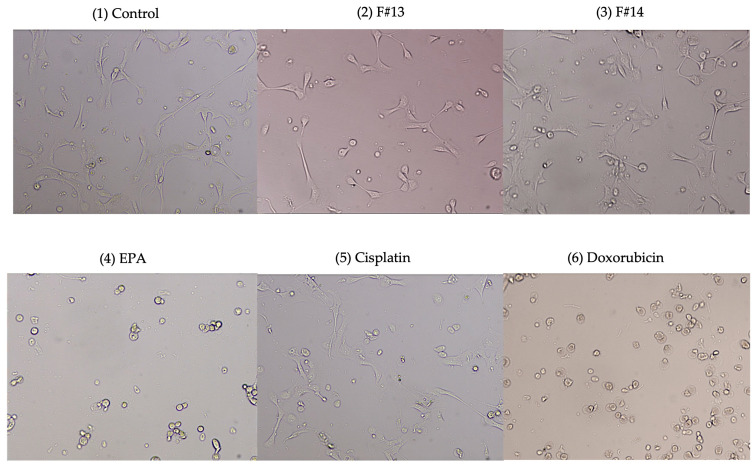
MCF-10A cells observed at 20×. Morphology after 48h treatment at 100 μg/mL.

**Figure 7 marinedrugs-23-00059-f007:**
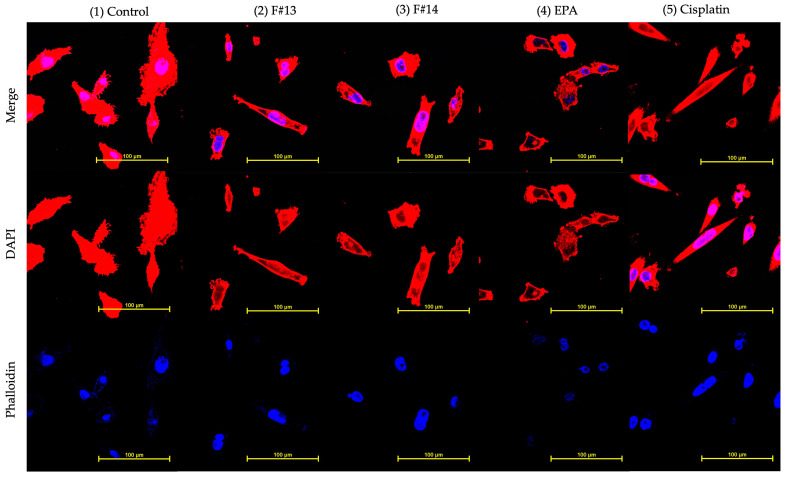
MDA-MB-231 cells observed at 60× after 24 h of treatment with (1) control, (2) F13 (IC_50_: 102 μg/mL), (3) F14 (IC_50_ 139 μg/mL), (4) EPA (IC_50_: 27 μg/mL), and (5) Cisplatin (IC_50_: 8 μg/mL). The cytoskeleton and DNA were dyed with phalloidin and DAPI, respectively.

**Table 1 marinedrugs-23-00059-t001:** Neutral and polar lipids present in chloroform extract and hexane and methanol partition from *L. vannamei* muscle.

Lipids	Chloroform	Hexane	Methanol
FFA (%)	16.49 ± 0.51	20.04 ± 3.47	15.11 ± 2.65
CHL (%)	26.32 ± 0.53	69.92 ± 4.1	16.77 ± 3.27
Total neutral lipids (%)	42.81	89.97	31.88
PE (%)	33.09 ± 1.23	7.49 ± 2.41	37.68 ± 2.13
PC (%)	24.09 ± 1.22	2.53 ± 0.81	30.43 ± 2.92
Total polar lipids(%)	57.19	10.02	68.11

Neutral lipids: Free fatty acids (FFA) and cholesterol (CHL). Polar lipids: Phosphatidylethanolamine (PE) and phosphatidylcholine (PC). The values are expressed in percentage of dry residue and are mean standard deviations of two different determinations.

**Table 2 marinedrugs-23-00059-t002:** Fatty acid profile of chloroform extract and methanol and hexane partition obtained from *L. vannamei* muscle.

A.G.	Chloroform (%)	Methanol (%)	Hexane (%)
14:0	0.31 ± 0.04	0.24 ± 0.0	0.51 ± 0.1
15:0	0.24 ± 0.02	0.25 ± 0.01	0.32 ± 0.04
16:0	14.54 ± 0.38	14.34 ± 0.34	16.35 ± 0.28
16:1n-9	0.11 ± 0.01	0.10 ± 0.0	0.15 ± 0.01
16:1n-7	0.76 ± 0.04	0.73 ± 0.03	1.18 ± 0.32
17:0	1.01 ± 0.06	1.06 ± 0.03	0.95 ± 0.12
16:2n-6	1.03 ± 0.0	1.09 ± 0.07	0.59 ± 0.01
16:2n-4	0.31 ± 0.0	0.28 ± 0.0	0.31 ± 0.0
18:0	12.48 ± 0.03	12.76 ± 0.14	11.19 ± 0.53
18:1n-9	12.47 ± 0.86	11.5 ± 0.27	20.34 ± 5.03
18:1n-7	2.38 ± 0.08	2.34 ± 0.01	2.58 ± 0.01
18:2n-6	17.11 ± 0.25	17.09 ± 0.07	17.42 ± 1.71
20:0	0.30 ± 0.02	0.32 ± 0.0	0.41 ± 0.02
18:3n-3	1.26 ± 0.04	1.25 ± 0.01	1.2 ± 0.04
20:1n-9	0.71 ± 0.07	0.62 ± 0.04	0.97 ± 0.09
18:4n-3	0.13 ± 0.02	0.11 ± 0.01	0.21 ± 0.06
20:2n-6	2.87 ± 0.03	2.92 ± 0.03	3.41 ± 0.06
20:3n-6	0.16 ± 0.01	0.16 ± 0.02	0.17 ± 0.07
22:0	0.27 ± 0.01	0.24 ± 0.01	0.38 ± 0.05
20:4n-6	2.55 ± 0.06	2.59 ± 0.06	2.13 ± 0.25
22:1n-9	0.21 ± 0.03	0.11 ± 0.02	0.12 ± 0.02
20:4n-3	0.22 ± 0.0	0.21 ± 0.03	0.19 ± 0.03
20:5n-3	12.83 ± 0.07	13.37 ± 0.02	7.84 ± 1.31
24:0	0.18 ± 0.01	0.25 ± 0.04	0.38 ± 0.01
22:4n-6	0.26 ± 0.01	0.26 ± 0.02	0.24 ± 0.1
24:1n-9	0.18 ± 0.03	0.14 ± 0.03	0.23 ± 0.01
22:5n-6	0.34 ± 0.01	0.36 ± 0.05	0.25 ± 0.03
22:5n-3	1.18 ± 0.03	1.19 ± 0.02	0.98 ± 0.08
22:6n-3	13.45 ± 0.02	14.04 ± 0.0	8.88 ± 1.42
SAT	29.49 ± 0.37	29.56 ± 0.20	30.63 ± 0.26
MUFA	16.83 ± 0.71	15.54 ± 0.24	25.57 ± 5.29
PUFA	53.68 ± 0.34	54.90 ± 0.03	43.8 ± 5.02
n-3	29.06 ± 0.1	30.17 ± 0.07	19.29 ± 2.94
n-6	24.31 ± 0.24	24.46 ± 0.04	24.2 ± 2.08
n-3/n-6	1.20 ± 0.01	1.23 ± 0.0	0.79 ± 0.05
EPA + DHA	26.28 ± 0.05	27.41 ± 0.02	16.72 ± 2.73
EPA/DHA	0.95 ± 0.01	0.95 ± 0.0	0.88 ± 0.01

The values are means ± standard deviation of fatty acid percent related to total fatty acid of two replicates of chloroform extract, methanol, and hexane partition.

**Table 3 marinedrugs-23-00059-t003:** Fatty acid profile of fractions with biological activity obtained from *L. vannamei muscle*.

Fatty Acids	% Fatty Acids F#13	% Fatty Acids F#14
14:0	0.61 ± 0.12	0.46 ± 0.01
15:0	0.52 ± 0.06	1.00 ± 0.02
16:0	9.17 ± 0.28	3.11 ± 0.09
16:1n-9	0.46 ± 0.00	0.41 ± 0.00
16:1n-7	1.07 ± 0.08	0.38 ± 0.08
17:0	0.40 ± 0.04	0.00 ± 0.00
16:2n-4	0.13 ± 0.01	0.00 ± 0.00
18:0	3.14 ± 0.08	1.34 ± 0.11
18:1n-9	7.96 ± 0.15	2.08 ± 0.18
18:1n-7	1.70 ± 0.01	0.00 ± 0.00
18:2n-6	26.21 ± 0.12	9.02 ± 0.08
20:0	0.17 ± 0.01	0.42 ± 0.01
18:3n-3	3.22 ± 0.05	2.51 ± 0.17
20:1n-9	0.28 ± 0.01	0.00 ± 0.00
21:0	0.11 ± 0.01	0.00 ± 0.00
20:2n-6	2.06 ± 0.03	0.74 ± 0.04
20:3n-6	0.22 ± 0.01	0.00 ± 0.00
22:0	1.22 ± 0.09	0.29 ± 0.08
20:4n-6	4.48 ± 0.1	3.09 ± 0.15
22:1n-9	0.14 ± 0.05	0.00 ± 0.00
20:4n-3	0.54 ± 0.05	0.51 ± 0.04
20:5n-3	21.40 ± 0.3	31.09 ± 0.38
24:0	0.87 ± 0.02	1.10 ± 0.09
22:4n-6	0.65 ± 0.02	0.64 ± 0.00
22:5n-6	0.40 ± 0.01	1.03 ± 0.10
22:5n-3	2.33 ± 0.05	1.43 ± 0.12
22:6n-3	10.32 ± 0.19	39.33 ± 0.25
SAT	16.20 ± 0.33	7.73 ± 0.18
MUFA	11.63 ± 0.35	2.87 ± 0.43
PUFA	72.17 ± 0.26	89.40 ± 0.39
n-3	37.81 ± 0.37	74.88 ± 0.50
n-6	34.21 ± 0.16	14.52 ± 0.20
n-3/n-6	1.10 ± 0.01	5.16 ± 0.12
EPA + DHA	31.72 ± 0.36	70.42 ± 0.45
EPA/DHA	2.07 ± 0.05	0.79 ± 0.02

The values are means ± standard deviation of fatty acids contained in F#13 and F#14 expressed as a percentage of fatty acids related to the total fatty acids of three replicates of each fraction.

**Table 4 marinedrugs-23-00059-t004:** Concentration (μM and μmol/ mg residue) of astaxanthin (AX + esters AX) and β-carotene in chloroform extract and hexane and methanol partition.

Extract	μM AX + Esters AX	μmol (AX + Esters AX)/mg Residue	μM β-Carotene	μmol β-Carotene/mg Residue
Chloroform	1.31	3.17 × 10^−4^	0.03	2.91 × 10^−5^
Hexane	1.35	6.26 × 10^−4^	0.40	2.15 × 10^−4^
Methanol	6.11	4.43 × 10^−3^	0.00	0.00 × 10^1^

AX (Astaxanthin).

**Table 5 marinedrugs-23-00059-t005:** Concentration pigment composition expressed as a percentage of pigment related to total pigments of chloroform, hexane, methanol, F13, and F14 extracts.

Compound	Chloroform (%)	Methanol (%)	Hexane (%)	F13 (%)	F14 (%)
Astaxanthin	51.98	48.36	6.93	70.88	<D. L.
Astaxanthin monoesters and diesters (EPA and DHA)	3.81	4.34	0.00	20.49	<D. L.
Astaxanthin diesters (EPA or DHA + SAT, MUFA or 18:2n-6)	41.84	47.30	66.80	8.63	<D. L.
β-carotene	2.37	0.00	26.28	0.00	0.00

<D. L.: Under limit detection.

**Table 6 marinedrugs-23-00059-t006:** Quantification of tocopherol from extracts and eluates obtained from *L. vannamei* muscle.

Tocopherol	Chloroform (%)	Methanol (%)	E 2 (%)	E 3 (%)
α-tocoferol	0.1789	0.1829	0.1112	0.0050
γ-tocoferol	0.0108	0.0074	0.0002	0.0057
δ-tocoferol	0.0898	0.0989	0.0807	0.0041

The values are expressed in percentage of mg of each tocopherol isomer related to residue in various *L. vannamei* muscle extracts.

**Table 7 marinedrugs-23-00059-t007:** Proportions of mobile phases for fractionation of the methanol phase obtained from shrimp muscle (*L. vannamei*) in open column chromatography.

	Solvent Percentage (%)	
Mobile Phase	Hexane	Acetone
1	95	5
2	90	10
3	85	15
4	80	20
5	75	25
6	70	30
7	65	35
8	60	40
9	55	45
10	50	50

## Data Availability

Data are available upon request.

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
