# Peer review of "Chemical Characterization of Bioactive Compounds in Extracts and Fractions from Litopenaeus vannamei Muscle"

_marinedrugs, 2025, doi:10.3390/md23020059_

Round 1
Reviewer 1 Report
Comments and Suggestions for Authors
See the attached.

Author Response
The manuscript entitled “Chemical Characterization of Bioactive Compounds in Extracts and Fractions from Litopenaeus vannamei Muscle” (marinedrugs-3406099) has been reviewed and corrected according to the editor and reviewer comments. We are very grateful for the comments and suggestions that have contributed to improve the manuscript. We have decided to resubmit this article to Marine Drugs addressing the comments made and doing a major revision accordingly.
Comments 1: The introduction section needs to more clearly state the research objectives and hypotheses.
Response: According to referee’s suggestion, we have been working during these last days to improve intensely the manuscript in order to clarify the research objectives and the hypotheses in the introduction section. All sections have been rewritten in order to improve the paper. Indeed and in agreement with referee number 2 comments, we have also carefully rewritten the abstract section to better draw the issue, the results achieved and the conclusions derived from the study. We have used the track tool to make easier the revision process done.
We have also reviewed carefully all the manuscript, references, tables and figure. An extensive revision of language, spelling, grammar and style has been done as well.
Comments 2: Page 4, Figure 3B, there were two “B”s in the figure.
Response: We have also reviewed carefully all the tables and figures. We have eliminated Figure 3B in order to make the manuscript clearer.
Comments 3: Page 5, line 133, “muscle” should not be italic, the authors should notice these details through the manuscript.
Response: All the text has been carefully reviewed to provide grammatical coherence with no spelling errors.
Comments 4: Page 8, Figure 5 and 6: The explanations in the figure captions are unclear. What do AB/A/B/C represent in the figures? The description of statistical significance is incorrect. It is stated that all data are statistically significant, which is clearly not the case.
Response: We apologize for any confusion caused by the statistical description of figures 4 and 5. We have replaced the letters AB/A/B/C with asterisks to indicate the data that are significantly different, and the description has been updated accordingly. We sincerely appreciate your valuable observation (Lines: 235-244).
Comments 5: Page 9, figure 6, the authors should add the linear scale in the figure.
Response: We deeply appreciate your valuable observation regarding the absence of the linear scale in the microscopy images. We are pleased to inform you that we have added the corresponding scale to figure 7. Unfortunately, it was not possible to add the scale to the images in Figure 6 due to an issue with the microscope software. However, we are committed to resolving this problem and ensuring that this important requirement is ready in case the manuscript is accepted. We appreciate your understanding and offer our sincere apology.
Comments 6: The bioassay was still in the surface; it is recommended that the authors conduct more in-depth mechanistic studies.
Response: We deeply appreciate your enriching comment. Tests such as flow cytometry, caspase activation assays, and western blot to identify proteins involved in antiproliferative processes can provide a greater understanding of the mechanisms of action. However, it has been considered to conduct these determinations in future research. This information is found in the final discussion section and the conclusions section (Lines: 328-334; 471-473).
Comments 7: The article needs to be carefully proofread to correct grammatical and spelling errors. The reference format also needs to be standardized.
Response: We have performed an extensive revision of language, spelling, grammar and style has been done as well.
Reference format has been carefully addressed
Trusting that you will find this manuscript amenable to your requirements on its present form, and waiting for your reply, I remain.
Kind regards,
Carolina De La Reé, Carmen López Saiz and Isabel Medina
Reviewer 2 Report
Comments and Suggestions for Authors
My opinion is: manuscript need major revision.
Abstract is not so informative, it describes mainly methods, but not results. In introduction no literature data on subject of investigation. Why authors used lipid extraction with only chloroform? I think that such extraction method need validation, because chloroform is not miscible with water. There is standard method of lipid isolation by Folch et al., 1957 and Bligh&Dyer, 1959, which give good and reproducible results.
1. In this paper were characterized only 2 phospholipids PE and PC. Why? Generally, in marine invertebrates presents phosphatidyl-choline (PC), ethanolamine (PE), serine (PS), inositol (PI), diphospatidyglycerol (DPG), and specifically for crustacean sphingomyelin (SM) [Vaskovsky V. Phospholipids. In Marine Biogenic Lipids, Fats and Oils. 1989. v.1. p.199].
2. At FAME analysis by GC did not specify temperatures column and detector. There is error in column specification (L × I.D. 30 m × 0.25 mm, df 0.20 μm, Supelco).
3 . Line 316. Correct please (150 x 4.6 mm, 3.5 m particle size, 100 Â pore size)
4. How were quantified phospholipids and neutral lipids? What means R1-R4 standard curve.
5. Line 218. What means saide fatty acids?
Author Response
The manuscript entitled “Chemical Characterization of Bioactive Compounds in Extracts and Fractions from Litopenaeus vannamei Muscle” (marinedrugs-3406099) has been reviewed and corrected according to the editor and reviewer comments. We are very grateful for the comments and suggestions that have contributed to improve the manuscript. We have decided to resubmit this article to Marine Drugs addressing the comments made and doing a major revision accordingly.
Comments 1: Abstract is no informative, it describes mainly methods but no results.
Response: As we mentioned above and according to referee’s suggestion, we have been working during these last days to improve intensely the manuscript in order to clarify the research objectives and the hypotheses in the introduction section. All sections have been rewritten in order to improve the paper. Indeed and in agreement with referee number 2 comments, we have also carefully rewritten the abstract section to better draw the issue, the results achieved and the conclusions derived from the study. We have used the track tool to make easier the revision process done.
Comments 2: In literature no literature data on subject of investigation
Response: We have also written the introduction including key references for helping to understand the subject and the hypothesis done.
Comments 3: Why authors used only chloroform for lipid extraction? I think that such extraction method needs validation, because chloroform is not miscible with water. There is standard method with lipid isolation by Folch et al 1957 and Bligh and Dyer 1959 which give good and reproducible results.
Response: We really appreciate referee’s suggestions related to accurate methods for lipid extraction. Indeed, we have clarified the manuscript in order to present clearly that the aim of the work is to study the properties of an organic chloroform extract which has already demonstrated to contain bioactive compounds (New reviewed Abstract and Introduction section). Then, the work is not aimed to perform a lipid extraction but an organic extraction. We have added a last paragraph in the introduction section describing clearly the objective of the work.
Then, this work is aimed to test the biological activity of marine organic extracts coming from L. vannamei muscle which have previously showed a certain antiproliferative activity over breast carcinoma cells and their innocuity over normal cells as breast epithelial cells. The study intents to deep inside the composition of these biologically active extracts confirming their specificity for breast adenocarcinoma cells and identifying the compounds responsible of the biological protective effect.
Comments 4: In this paper were characterised only two phospholipids, PE and PC. Why? In general in marine invertebrates presents phosphatidylcoline (PC), ethanolamine (PE), serine (PS), inositol (PI), diphosphatidylglycerol (DPG), and especially in crustaceans sphingomyelin (SPM). Vaskosky V, Phospholipids in Marine Biogenic Lipids, Fats and Oils, 1989. V1. P.199.
Response: Again, we thank referee for his explanation and help. In fact, total lipid extract of crustaceans contained PC, PE, PS, PI, DPG and SPM. However, as we mentioned above, here we are not working with the total lipid extract. We are not working with a phospholipid extract either, but in an organic extract coming from chloroform and washed in a mixture of methanol and hexane. Then, we have rewritten the introduction and the results section to clarify that the phospholipids detected, PC and PE, are detected in the isolated organic extract coming from the procedure. Such is, a first extraction with chloroform, followed by a second extraction on a mixture of methanol and hexane. This methanol extract contained only PC and PE.
Comments 5: At FAME analysis by GC did not specify temperatures of column and detector. There is an error in column specification (L x id 30 m x 0.25 mm, df 0.20 Supelco).
Response: We apologize for this mistake that has been currently corrected. Additionally, we have now included the data related to temperatures of detector and column (Line 394).
Comments 6: Line 316. Correct please 150 x 4.6 mm, 3.5 m particle size, 100 A pore size.
Response: We apologize for this mistake that has been currently corrected (Line 403).
Comments 7: How were the phospholipids and neutral lipids quantified? What means R1-R4 standard curve.
Response: Phospholipids and neutral lipids were quantified using a densitometric procedure, taking references from a standard curve made by lipid standards employed in concentrations ranged from low levels to higher levels. The methodology section has been rewritten to give details about this quantification (Lines 365-385).
Neutral lipid classes: Quantification was performed using standard calibration graphs having concentration values (R1-R4) ranged between: Cholesterol (CHL): 2,69- 16, 16 ug/ul; Free fatty acids (FFA): 2,69 - 13,76 ug/u; Triacyl glycerides (TG): 0,54-5,38 ug/uL; Diacylglycerides (DG): 0,56-5,59 ug/uL and Monoacylglycerides (MG): 0,44- 4,40 ug/uL.
Polar lipid classes: Quantification was performed using standard calibration graphs having concentration values (R1-R4) ranged between: Phosphatidylcholine (PC): 0,73-6,35 ug/uL and Phosphatidylethanolamine (PE): 1,41-8,78 ug/uL.
Comments 8: Line 218. What means said fatty acids?
Response: We apologize for the mistake. The right word is unsaturated fatty acids (Line: 288)
Trusting that you will find this manuscript amenable to your requirements on its present form, and waiting for your reply, I remain.
Kind regards,
Carolina De La Reé, Carmen López Saiz and Isabel Medina
Round 2
Reviewer 1 Report
Comments and Suggestions for Authors
Accept in present form
Author Response
Comment: Accept in present form
Response: We sincerely appreciate your comments aimed at improving this manuscript.
Reviewer 2 Report
Comments and Suggestions for Authors
Authors positively responded on most my comments, beside comment 3 and 4. Obviously, if someone works with lipids, he must follow lipid methods. Otherwise the results can be not full. I think that this manuscript has low significance. I will agree with Academic Editor's desition .
Author Response
Comments: Authors positively responded on most my comments, beside comment 3 and 4. Obviously, if someone works with lipids, he must follow lipid methods. Otherwise the results can be not full. I think that this manuscript has low significance. I will agree with Academic Editor's desition .
Response: We sincerely appreciate your comments aimed at improving this manuscript. We have worked on the manuscript to enhance it.
Round 3
Reviewer 2 Report
Comments and Suggestions for Authors
I have some additional notes.
In fatty acid abbreviations more correct is use w instead of “w”.
How many samples were analyzed in Table 3? Is it real sensitivity up to 0.001%.
In Reference 27 (Lepage and Roy, 1986) authors used for methylation MeOH-acethyl chloride. In this MS Line 404 mentioned sulfuric acid.
Author Response
|
The manuscript entitled “Chemical Characterization of Bioactive Compounds in Extracts and Fractions from Litopenaeus vannamei Muscle” (marinedrugs-3406099) has been reviewed and corrected according to the reviewers and Editor comments. We are very grateful for the comments and suggestions that have contributed to improve the manuscript. We have decided to resubmit this article to Marine Drugs addressing the comments made and doing a major revision accordingly.
Trusting that you will find this manuscript amenable to your requirements on its present form, and waiting for your reply, I remain. Kind regards, Carolina De La Reé, Carmen López Saiz and Isabel Medina |
